# Bone Abrasive Machining: Influence of Tool Geometry and Cortical Bone Anisotropic Structure on Crack Propagation

**DOI:** 10.3390/jfb13030154

**Published:** 2022-09-15

**Authors:** Paweł Zawadzki, Rafał Talar

**Affiliations:** Faculty of Mechanical Engineering, Poznan University of Technology, Maria Sklodowska-Curie Square 5, 60-965 Poznan, Poland

**Keywords:** cortical bone, orthogonal cutting, bone fracture mechanism, anisotropic bone properties, crack propagation, abrasive machining

## Abstract

The abrasive machining of cortical tissue is used in many arthroplasties and craniofacial surgery procedures. However, this method requires further research due to the processes’ complexity and the tissue’s composite structure. Therefore, studies were carried out to assess the impact of grid geometry and the anisotropic structure of bone tissue on the cutting process and crack propagation. The analysis was performed based on an orthogonal cutting in three directions. The grain shape has been simplified, and the cutting forces, crack path and surface quality were monitored. The results indicate that a depth of cut at 100–25 µm allows the most accurate cutting control. A transverse cutting direction results in the greatest surface irregularity: *Iz* = 17.7%, *Vvc* = 3.29 mL/m^2^ and *d_f_* = 5.22 µm and generates the most uncontrolled cracks. Maximum fracture force values of *FF* > 80 N were generated for *d* = 175 µm. For *d* < 5 µm, no cracks or only slight penetration occurs. A positive *γ* provides greater repeatability and crack control. Negative *γ* generates penetrating cracks and uncontrolled material damage. The individual types of cracks have a characteristic course of changes in *F_x_*. The clearance angle did not affect the crack propagation.

## 1. Introduction

Cortical bone tissue machining is one of orthopodic surgery’s most frequently used procedures. This method is highly invasive [1] and used primarily during hip and knee arthroplasty [2], limb fracture surgery, spine arthroplasty and other joint replacement procedures [3]. One of the methods used during orthopedic procedures is cartilage and bone tissue surface machining based on drilling, cutting and milling [4]. Unfortunately, all types of mechanical impact involve thermal energy emission [5], tissue destruction and the extension of convalescence time [6]. Moreover, the popularity of standardized surgical procedures causes no changes in the processing technology and the availability of specialized surgical equipment.

In his research, the author proposes the use of abrasive machining in order to shape the surface of compact bone tissue. It provides the possibility of shaping complex geometries. However, abrasive machining involves a complex material cutting process. One of the critical research issues is the characterization of the fracture process and the propagation of the fractures of the bone tissue compacted during cutting. Many scientists have carried out fundamental research on this process. Bonfield [7], using various techniques including the single edge notched, centre notched cylindrical and compact tension methods, has shown that the stress intensity factor and the critical strain energy release rate obtained depend on the orientation of the cortical bone, as well as on bone density, the velocity of crack propagation and specimen geometry. In other studies, Bingbing et al. [8], using a combination of experimental and numerical approaches, showed that the resistance to transverse fractures was significantly higher compared to the parallel ones, indicating that the cortical properties of the fractures are essentially anisotropic. As the fracture grew in the transverse direction, the fracture deflected significantly, and branching was found in the fracture trace, while in the longitudinal direction, the fracture was straight, and no ramification was observed. Extensive research on the propagation of cracks in cortical bone was carried out by Gustafsson et al. [9,10].

A comprehensive study of material parameters was performed using a 2D XFEM failure model to simulate crack propagation around an osteon at a micro-scale. The results identified factors related to the cement line to influence crack propagation, wherein the interface strength was necessary for the ability to deflect cracks [10]. Weak cement line interfaces changed the orientation of propagating cracks, while models with robust interfaces predicted crack trajectories that penetrated the cement line and propagated through the osteons [9].

The studies presented above did not analyze the case in which the material is cut with a specific cutting tool, and its “cracking” is a deliberate procedure. It should be noted that bone tissue is a complex material that includes both microstructure elements (osteons with a diameter of 100–300 μm) and submicron structures (bone plates 3–7 μm in size). The cortical bone is a quasi-fragile living tissue, and these properties allow the easy propagation of cracks in specific directions of the structure. Failure to control fractures can damage the blood vessel space and nerve endings. Therefore, for the sake of the patient’s health, special attention should be paid to the mechanism of cutting the bone tissue.

In abrasive machining, a group of grains with a generally undefined geometry cuts the tissue. In order to characterize the fracture process, it is necessary to simplify the geometry of the cutting tools through its parameterization and to indicate the test method. The commonly used orthogonal cutting with the use of tools with a defined cutting geometry was adopted as the method. So far, the most thorough research in this area has been carried out by Liao et al. [11] and Bai et al. [12]. Liao et al. [11] used one tool with rake angle *γ* = 8° and clearance angle *α* = 8°. The same choice was used by Liao et al. [10] to simulate bone milling processes. Similarly, one tool was used by Bai et al., [12] with rake angle *γ* = 10° and clearance angle *α* = 7°. Jacobs et al. [13] used a more comprehensive range of tools in pioneering research using rake angle *γ* = −5, 0, 15, 35 and 45°; clearance angle was not provided. Wiggins et al. [14] also introduced a more comprehensive range of angles by taking rake angle *γ* = −10°, 10° and 40° and clearance angle *α* = 10°. The results of all studies showed the influence of the anisotropic bone structure on the character of the results, although the narrow range of the depth of the cut and the geometry of the cutting tools used in the above studies do not make it possible to understand the complexity of the entire process.

Wu et al. [15] used the differential quadrature hierarchical finite element method (DQHFEM) to analyze the isotropic and composite structures. This model shows numerical instability and is less sensitive to locking effects and geometry distortion. This model shows a high potential for applying 3D composite structure modeling. The model based on DQFEM and on improved hierarchical Legendre expansion for composite materials was also prepared by Yan et al. [16]. The applied method reduces the computational cost relative to conventional solutions. Both of the above solutions are used in simulation models of bone tissue. Solutions based on Burnstein polynomial basis functions and an accurate Bézier-based multi-step method were developed and implemented by Kabir et al. [17]. Comparing the numerical and analytical results indicates the stability of the solution. The method correctly indicates the necessity of introducing structural changes in the properties of composites. Among others, simulation research on scaffolds was conducted by Hashemi et al. [18], who showed that an increase in porosity increases tensions in the scaffolding structure. Therefore, special attention should be paid to the porosity of the composite structure.

The following work describes the nature of the fracture of compacted bone tissue, depending on the cutting tool’s geometry and the cut’s depth. Particular attention was paid to the microstructural features and anisotropy of the cortical bone tissue. The tissue fracture process was observed and analyzed during transverse cutting, parallel and across. The cutting forces were measured, and both the fracture process and the crack propagation were recorded using a microscope camera. Measurements of volumetric parameters of surface topography were also performed. For the first time, an accurate analysis of the cutting of compacted bone tissue in terms of fracture evaluation was obtained.

## 2. Materials and Methods

This study included the fundamental technological effects of the orthogonal cutting of cortical bone tissue using a cutting tool with a defined geometry. Cutting forces were measured during the experiments, and a microscope camera recorded the fracture mechanism. Based on the recorded data, crack exciting force, crack propagation path, crack size and overall characteristics were considered. An orthogonal cutting process was chosen, often used in basic research and industry. Cortical bone tissue is the basic building block of the human skeletal system; thus, orthopodic surgeons often operate on the material. It is characterized by approximately 5–10% porosity [19]. From a mechanical point of view, cortical bone carries a large part of the load (from 30 to 90% of the axial loads on long bones) [20,21,22,23]. Porosity influenced the mechanical tensile and compressive strength [24]. Bone porosity is negatively related to the modulus of elasticity, hardness and elasticity [25]. The compression cracks close, increasing compressive strength and tensile strength. The variable porosity of the tissue causes stress concentration, which serves as a stimulus for crack initiation, propagating as microcracks [26].

Therefore, cortical bone tissue was selected to preserve the most homogeneous structure. It ensures the stability and repeatability of experimental studies. The prepared complex set of studies has allowed the broadest research on cutting bone tissue machining.

### 2.1. Bone Characteristics

Cortical tissue is a heterogeneous composite material with elements including hydroxyapatite mineral (Ca10 (PO4) 6 (OH) 2) [27], a mixed organic component and water [28,29]. Measurements of the bone tissue composition using the XRF method show the following chemical composition of the bone tissue: Ca-72.3%, P-26.68%, K-0.78% and others < 0.3% [30]. The fundamental structural elements of cortical bone tissue are osteons. The shape of elliptical osteon cylinders is a diameter of 100–300 µm and a length of 3–5 mm [31,32,33]. The osteons surround the Haversian canals and are separated from the interstitial bone tissue by a thin layer of amorphous substance deficient in collagen, called a cement line (0.5 to 1 μm). Each osteon is made of concentric lamellae (1 to 5 μm thick), among which the bone cells reside inside ellipsoidal spaces (10 to 50 μm). Cortical bone tissue differs from cancellous bone tissue in the content of calcium, phosphorus, water, density and mineralization level [34,35]. The elastic modulus in the longitudinal direction (17.4 GPa for humans and 20.4 GPa for bovine) is more significant than that in the transverse (9.6 GPa for human bone and 11.7 GPa for bovine bone) [36]. Research studies suggest that bone tissue should be cut in three directions [11,12,13,14]. Anisotropic properties were adopted following the assumptions of Cowin and Sadegh [37]. The results of studies by Abdel-Wahab et al. [38] and Sugita et al. [39] indicate that the interstitial matrix has a slightly higher elastic modulus (22.8 GPa) than cortical bone. The cemented line is characterized, respectively, by 6.85 GPa and 39 MPa. The compressive strength of bone is 100–130 MPa [40].

### 2.2. Bone Specimens

Fresh cattle femurs were obtained from a local slaughterhouse and kept in Ringer’s fluid. The bones came from healthy, mature animals. Samples were taken from the central shaft. Literature reports confirm that the mechanical properties are similar to the human cortical bone [41]. The sizes of the samples were 20 mm × 4 mm × 10 mm. Due to the anisotropic structure of the cortical tissue, it was necessary to collect samples with three independent orientations [42] (see Figure 1). The samples were polished with 320 grit sandpaper, dipped in Ringer’s liquid and stored in a refrigerator at 0 °C (after testing at −25 °C).

According to Kaye et al. [43] frozen bone samples can be used in studies focused on the mechanical properties of bone. The literature does not show any differences from the frozen to the fresh cortical bone in terms of mechanical properties [44,45].

### 2.3. Cutting Tools Geometry

In order to work out the influence of the geometry of the abrasive grain cutting edge on the cutting process, the following simplification was adopted. It was assumed that the abrasive tool consists of a set of grains of undefined geometry, although their shape can be generalized by adopting a wide range of rake and clearance angles. It is possible to analyze a wide range of cutting tools with a specific geometry, allowing one to recreate the actual state (see Figure 2). Therefore, a set of 27 cutting tools with a specific cutting geometry was prepared. The determined cutting geometry was the rake angle, the clearance angle and the 10 mm cutting edge width. The variability of the rake and relief angles reflect the wide range of abrasive grain geometries in grinding wheels. The values of the tool rake angles γ and the clearance angles α are presented in Table 1.

High-speed steel cutting tools were manufactured by Avanti-Tools Sp. Z OO (Poznań, Poland) as designed by the author. An example of cutting tools is shown in Figure 3. The width of the tool’s cutting edge was 10 mm. The tools were mounted in a specialized holder made of structural steel (see Figure 3E). It should be noted that such a wide range of tool geometries (27 cutting tools were prepared) has not been analyzed yet (see Table 1).

### 2.4. Experimental Setup

The measuring system was developed to study the orthogonal cutting process of cortical bone tissue. The tool movement was performed by UMT Bruker tribotester drives equipped with a 3-axis motion system and high-resolution stepper motors (see Figure 3). The DFM−20 two-axis force sensor measured the force with a range of 0.05 to 235 N. The following features characterize it: measuring resolution of 0.01 N, non-linearity of 0.02% and sampling frequency of 1000 Hz. It provides precise measurement of force and position in three axes. The Motic optical microscope with a microscope camera with image registration was used to monitor the cutting process. The camera has a maximum resolution of 2048 × 1536 pixels. Pictures were taken during the fracture propagation process.

Table 1 shows the implemented input parameters. The wide range of tool geometries reflected the geometry of the abrasive grains. The cutting depths are based on two characteristics. The first is a variable value of the depth of cut depending on the geometric parameters of a single grain. The second is the material cut type (lamellar or osteon). The range from 0.5 to 50 µm covers the thickness of the lamellar, and higher values from 50 to 175 µm cover the dimension of the osteon diameter. A constant cutting speed *v_c_* = 30 mm/min was established to minimize the influence of thermal effects on the cutting process. Research shows that below this value, there is no thermal necrosis [11].

As a result of the measurements, the tangential force *F_x_* and the contact force *F_y_* were obtained. Figure 4 shows a cross-section of the orthogonal cutting process. The device measured the direct value of the force *F_x_* and *F_y_*. The constant control of the position ensured a constant value of the depth of cut *d*. The value of the rake angle, clearance angle and corner radius *r* depended on the tool used. In the analyzed cases, attention was focused on the value of the force *F_x_* being a direct factor of the generated cracks.

Surface topography analyses were performed on a contact profilometer. The functional surface analysis is based on volume parameters: material peak volume *Vmp*, material core volume *Vmc*, core void volume *Vvc* and cavity void volume *Vvv*. The parameters mentioned are expressed in mL/mm^2^ and are calculated from the material ratio curve.

## 3. Results

Three characteristic cases (*γ* = 40°, 0° and −40°) for depths *d* = 175, 50 and 5 μm are described to illustrate the evaluation methodology and characterize the cutting process. Conclusions for all cutting parameters and depth are presented in Table 2. The clearance angle *α* did not affect the obtained results; therefore, it was not commented on in the following parts of this chapter.

### 3.1. d = 175 μm

Figure 5 refers to a cutting depth of 175 μm. For *γ* = 40°, the most significant *F_x_* values in the transverse direction were observed, with high increases from *F_x_*_max_ to *F_x_*_min_ = 0. It indicates an apparent crack perpendicular to the direction of movement (parallel to the axis of osteones), with an immediate transition to cutting. There is an accumulation of a high amount of energy, which leads to a sudden fracture. Sharp cracks are also noticeable (see Figure 6A). For transverse movement, the rarer *F_x_* increases with lower observed values, characterized by reaching *F_x_*_max_ values and then falling and maintaining *F_x_*_min_ = 0 or *F_x_*_min_ = const. (see Figure 5). It indicates a rapid crack along the feed direction at the set speed level, leading to a short-term lack of contact at the tool’s top with the cut tissue. For parallel movement, cracks are characterized by considerable and minor *F_x_* increases of a transitional nature between the previous directions. A more significant number of cracks is noticeable, with lower fluctuations in *F_x_*. *F_x_*_min_ values are always greater than 0.

Bai et al. [12] obtained similar results for the directions of transverse and across, although it is difficult to demonstrate that cracks include only the spaces between osteones for parallel motion. The propagation of cracks along the cutting direction certainly has an indicated tendency, although the compressed material’s high pressure causes cracks to be generated through osteones. Similarly, Liao et al. [11] indicate that, in the parallel direction, the propagation of cracks does not have to occur only in the direction parallel to the feed and can change as a result of cracking cement spaces. This process was called the “peel up” of osteons. However, it should be noted that the compression of the material is not considered. For the directions of transverse and across, Liao et al. [11] presented similar observations.

It should be noted that osteones are not uniform, a fact that results in irregular cracks. In Figure 6, cracks along the cutting direction (HC) are noticeable, passing into transverse cracks (TC), perpendicular to the cutting direction. In the case of the transverse direction, cracks propagate in the interstitial matrix spaces; for the across and parallel directions, the indicated tendencies are not noticed. For the parallel direction, cracks occur more often, forced by the stresses generated by the deflection of the osteones.

For *γ* = 0°, in the case of transverse movement, regular increases between *F_x_*_max_ and *F_x_*_min_ are visible at a regular amplitude (see Figure 7). Across-cutting is characterized by lower *F_x_* values and lower amplitudes of value changes. It indicates shear and transverse cracks to the direction of movement, of small depth, with more extensive chip lengths. The most significant irregularity characterizes parallel movement. The most common changes in *F_x_* values occur with irregular amplitude fluctuations.

It should be noted that *F_x_*_min_ does not take the value 0. It indicates the constant resistance of the material; its partial accumulation in front of the tool face; and continuous, fine chip cutting. There is a compression of the material in all three cases, which breaks “shells out” uncontrollably. Of the previous studies, only Jacobs et al. [13] addressed the issue of rake angle *γ* = 0°, although the presented results did not allow for a comparison. The cracks presented in Figure 8 are initially longitudinal to the cutting direction and then go to the transverse cracks oriented at an angle close to 45° relative to the movement of the tool. This is due to the high pressure on the cortical bone. The fracture occurs because of cutting the structure and not because of cutting the tool and “breaking”. Cracks, in all cases, are irregular.

For *γ* = −40°, the most significant *F_x_* values in the transverse direction, with small but sudden increases from *F_x_*_max_ to *F_x_*_min_ > 0 are observed (see Figure 9). This indicates an apparent perpendicular fracture of the tissue to the direction of movement (parallel to the axis of osteones), with an immediate transition to cutting. The cracks are shallow, reducing the movement to “slipping”. *F_x_* increases are rarer with a higher amplitude across the movement. However, the cracks are immediate, and the processed material resists. It indicates a rapid crack perpendicular to the feed direction. For movement, parallel cracks are characterized by considerable *F_x_* increases with high amplitude, and cracks occur perpendicular to the direction of movement. A more significant number of cracks is noticeable, with high fluctuations in *F_x_*_max_. In the case of tools with negative *γ* < 0 angles, there is a clear tendency to resist the material constantly and high *F_x_*_min_ > 0. This may indicate the accumulation of material in front of the tool’s cutting edge. *F_x_*_min_ > 0 indicates that the tool constantly exerts pressure, so that the chips do not break off parallel to the tool’s edge. The cutting edge is in contact with the material all of the time. The cracks are irregular, and penetration into the tissue due to high pressure occurs. Only cracks in the interstitial matrix or through osteones were not noticed.

A shear crack is visible in Figure 10A. Material fracture is visible, and slight deformation is observed, which indicates the presence of brittle cracks. Brittle fractures display either transgranular or intergranular fractures. The crack travels along the grain boundaries and not through the grains themselves, and this usually happens when the grain boundary is weak and brittle. It is probably an example of intercrystalline fracture.

### 3.2. d = 50 μm

Figure 11 refers to a cutting depth of 50 μm. For *γ* = 40°, the largest *F_x_* values were obtained in the transverse direction, with high increases from *F_xmax_* to *F_xmin_*. It indicates a clear perpendicular fracture of the tissue to the direction of movement (parallel with the axis of osteones), with an immediate transition to cutting the following fragment. Compared to the depth of *d* = 175 μm, *F_x_*_min_ = 0 is not achieved. It is evidence of the formation of continuous, uninterrupted chips. There are small microcracks along the axis of osteones. As Bai et al. [12] noted, the cracks visible at the lateral edge of the chip are characterized by a spacing that, through their repeatability, defuses cemented lines in bone tissue. More frequent *F_x_* with lower values, characterized by more pronounced regularity, were obtained for the across movement. Compared to d = 175 μm, *F_x_*_min_ = 0 and *F_x_*_min_ = const were not received. This indicates a rapid crack perpendicular to the feed direction, forming chips at the set speed level. The cracks are shorter and do not cause the chip to break, and the cutting edge has constant contact with bone tissue. Increases from *F_x_*_max_ to *F_x_*_min_ > 0 indicate cracks in the chip structure while maintaining tightness.

For parallel movement, cracks are characterized by numerous minor *F_x_* increases of a transitional nature between the previous directions. A more significant number of cracks is noticeable, with lower fluctuations in *F_x_*. *F_x_*_min_ values are always greater than 0. The material breaks into the osteones’ axis length, peels it from the composite structure or cuts with the tool longitudinally. The process is characterized by high regularity, as well as stealth. The most favorable chip quality is preserved in this case, with minor transverse cracks. The presented results are consistent with the observations of Bai et al. [12] but differ from those presented by Liao et al. [11], who indicated that, in all directions, the formation of cracks interrupt the continuity of the chip, reducing it to a fragmentary form. The difference is probably due to the rake angle *γ* = 8°. In all directions, the chips take on similar shapes (see Figure 12).

Due to the brittle properties of bone tissue for lower cutting depths, low tensile strength in which a crack can easily be experienced compared to metals dominates.

For *γ* = 0°, in the transverse direction, a similar waveform irregularity was obtained as in the case of *γ* = 40°. Evident changes are noticeable for the *F_x_* value of across and parallel waveforms—the average value increases (Figure 13). For parallel runs, irregularity also increases. There is an increased accumulation of stresses and then the propagation of cracks transversely to the movement of the tool. These deep cracks result in tearing out part of the material structure (Figure 14). We obtain twice the *F_x_* value across traffic with decreasing amplitudes of changes. Cracks are noticeable along the cutting direction, with no transverse cracks for across movement, transverse cracks outside the cutting zone for the parallel direction or pronounced transverse cracks for transverse movement (Figure 13).

For *γ* = −40°, an apparent stabilization of the *F_x_* value for all directions in the transverse direction was obtained. For transverse traffic, low amplitudes of *F_x_* changes are noticeable, with an increasing load in absolute terms, which may indicate the accumulation of stresses in the surface being cut. Cracks are frequent, although chips are constantly formed, without cracks along the cutting direction (see Figure 15). In the parallel direction, stable chip formation occurs at transverse cracks. The highest amplitudes of *F_x_* are noticeable across the movement, but *F_x_*_min_ > 0. In comparison with the results obtained for *γ* = 40°, it should be noted that the process is stabilized while increasing the average value of *F_x_*. First, there is a partial “scraping” of the surface without the tool delving into the material. The material is crushed (see Figure 16), and the cracks diverge at an angle of 45°.

### 3.3. d = 5 μm

Cutting at a depth of *d* = 5 μm is characterized by the highest regularity of the course of changes while maintaining the high stabilization of *F_x_* values (see Figure 17). For *γ* = 40°, the course of changes in the across and transverse directions is similar. It is characterized by low *F_x_* values and low amplitude of lesions. It indicates numerous cracks in the structure of bone tissue at a low depth. For the parallel direction, higher *F_x_* values and fluctuations in the force amplitude are noticeable. This may indicate the encountering of osteones during cutting, the structure of which resists increasing the cutting force. Cracks were noted along the cutting direction. The characteristics are similar in all directions (see Figure 18).

The tendency to smooth the course of *F_x_* changes is noticeable already at *γ* = 0°. The *F_x_* values for the parallel direction dominate over the two whose values are similar. Cutting in the parallel direction is irregular. Frequent growths and *F_x_* flows can cause longitudinal cracking and the formation of shallow but long chips. Cracks are irregular. The across and transverse directions are characterized by similar waveforms. Penetrating the material at a low depth leads to transverse cracks, often of low *F_x_* amplitude.

For negative angles, *γ* = −40°, an apparent density of *F_x_* amplitude changes on the graph with a decrease in the differences between *F_x_*_max_ and *F_x_*_min_ observed (see Figure 19). The parallel direction is clearly distinguished from the others, with the characteristics of its course being similar. In all cases, mixed cracks (transverse and longitudinal) are noticeable, and the material is partially “scraped off” from the surface (see Figure 20).

For negative angles, *γ* = −40°, an apparent density of *F_x_* amplitude changes on the graph with a decrease in the differences between *F_x_*_max_ and *F_x_*_min_ were observed. The parallel direction is clearly distinguished from the others, with the characteristics of its course being similar (see Figure 21). In all cases, mixed cracks (transverse and longitudinal) are noticeable, and the material is partially “scraped off” from the surface (see Figure 22).

### 3.4. Surface Morphology

The result of cracks generated by cutting in a specific direction is the propagation of damage affecting the surface morphology. Analyses show a correlation between the cutting direction and the surface roughness, especially its volumetric parameters. Figure 23 shows fragments of the *F_x_* changes as a function of distance *l*. A lower frequency of changes in *F_x_* in the case of cutting in the direction parallel to the orientation of the osteons is noticeable. This difference is confirmed by the surface morphology parameters presented in Figure 24 and Table 3.

The porosity is closed as a side effect when porous materials are processed. The finishing process significantly reduces the surface porosity, reducing the risk of fatigue damage propagating from the surface. On a macroscopic scale, porosity affects cutting resistance values and chip characteristics. There is noticeable cohesion between the size of the cutting force and the size of the pores. On the microscopic scale, intermittent cutting can occur using fine-edged tools [46].

The 2D representation (see Figure 24) indicates the greater complexity of the surface for cuts in the transverse and across directions. Surface imperfections for the across direction assume the horizontal orientation of the arrangement (see Figure 24B), and for the transverse direction, they are point-like (see Figure 24A). The analysis of volumetric parameters shows the highest mean furrow depth *d_f_* (see Table 2) for the transverse direction, as well as the surface anisotropy *Iz* = 17.7%.

The volume of the valley voids, *Vvv*, results in the smallest depletion of material per unit area for the material fraction for the parallel direction. The volume of the core voids, *Vvc*, indicates the presence of numerous depressions on the surface, especially in the transverse direction, and it is probably related to the cracks’ intensity (see Figure 6B). The difference in the value of *Vmp*, that is, the volume of the elevation material, is not significant. *Vmc*, similarly to other cases, indicates that the hills are filled with the material to a greater extent than the valleys, which again proves the presence of great damage directed into the material. It is also confirmed by the *Rq* parameter, which indicates the profile’s most significant mean square deviation for the transverse direction.

## 4. Discussion

Few researchers have described the process of cortical bone chip formation and crack propagation. Liao et al. [11] used one tool with rake angle *γ* = 8° and clearance angle *α* = 8°. The same choice was used by Liao et al. [10] to simulate bone milling processes. Similarly, one tool was used by Bai et al. [12] with rake angle *γ* = 10° and clearance angle *α* = 7°. Jacobs et al. [13] used a more comprehensive range of tools in pioneering research using rake angle *γ* = −5, 0, 15, 35 and 45°; clearance angle was not provided. Wiggins et al. [14] also introduced a more comprehensive range of angles by taking rake angle *γ* = −10°, 10° and 40° and clearance angle *α* = 10°. the contact of the tool with bone tissue resulted in chip formation, and during this process, data on the formation and morphology of chips were recorded. As in the case of the Bai et al. [12] study, the results were divided into three cutting directions, presented in Figure 1. The course of *F_x_* values relative to the feed *l* was analyzed. These results allow conclusions about the nature of the cracks, depth, density and chip morphology.

The analysis of the fracture mechanism indicates significant differences in the process of cutting the material depending on the depth of cut and the rake angles. First, as the depth increases, the value of the cutting force *F_x_* increases. The correlation coefficient in all runs takes the value of *ρ* > 0.90. It indicates an increasing resistance of the material, which results in the formation of evident cracks, the splitting of the material, the breaking of the material, and breaking for large depths d > 125. High Δ*F_c_* is obtained. Stresses accumulate in the contact zone of the top of the tool, leading to discharge due to uncontrolled fracture. For depths in the range of *d* = 100–10 passes, the process is regular and becomes entirely regular at *d* < 10. Sudden decreases and increases in *F_x_* values indicate cracks in the material structure perpendicular to the direction of the movement of the tool (see Figure 5). Horizontal cracks result in the plucking of longitudinal pieces of material and are noticeable when the *F_x_* value drops to 0 or remains constant. The highest *F_x_* values for the range d > 10 are obtained for transverse traffic. For the range *d* < 10, there is a change in the parallel direction that generates higher loads. In the case of force *F_y_* directed perpendicular to the feed direction, a tendency for its regular increase was noted only for *γ* ≤ 0° (correlation coefficient *ρ* > 0.90). There are irregular changes for values of *γ* > 0° as the depth increases. This is probably the result of the tool’s delving into the tissue, resulting in increased resistance. The tendency to increase the value of *F_y_* with the increase in the negative cutting angle should also be emphasized. Large negative approach angles usually lead to higher cutting forces in the machining processes indicated by Akbari et al. [47] and Aurich et al. [48], concluding that machining with large negative rake angles induces high compressive loads and then generates high temperatures. These observations were also noted by Fang et al. [49] indicating the influence of the negative rake angle on the *F_x_*/*F_t_* force ratio. The increase in the *F_y_* force indicates the elastic deformation of the tissue, denting it under the tool, favored by the cutting angle’s negative orientation.

Among the types of cracks of the bone structure, those directed to the movement of the tool predominate. Longitudinal and angularly oriented cracks also appear. As in the case of Liao et al. [6]’s research, the cutting process models can be distinguished: shear cutting mode, shear-crack cutting and fracture cutting. However, shear cutting should be characterized as crushing shear for negative rake angles. The process results in chip fragmentation, reduced cutting efficiency, increased friction and increased temperature (Table 3). Fluctuations of forces noticeable on all waveforms allow the characterization of the process of cutting tissue. The most significant fluctuations are noticeable for high values of d and *γ* > 0, and a decrease in d and *γ* < 0 reduces the amplitude of *F_x_* fluctuations. The parallel cutting direction is characterized by stability, which was also shown by Bai et al. [12] or Zhu et al. [40], who analyzed cutting wood with a composite structure similar to cortical bone tissue. Fluctuations could indicate what type of chip can be expected during machining.

The force value is partially correlated with the type of cracks formed. High peaks characterize cracks for *d* > 100 μm (see *FF* in Figure 5). This is due to the resistance of the material. An increased frequency of *F_x_* changes accompanies cracks of this type. For transverse cracks (T), sudden drops in the value of *F_x_* are noticeable, with *F_xmin_* ≈ 0 (see Figure 5). Horizontal cracks (H) are characterized by assuming lower *Fx* values concerning transverse cracks, and the amplitude of F_x_ changes also decreases and their frequency increases. At the moment of the formation of a shear (S) crack, *F_xmax_* is reached, and then the *F_x_* value fluctuates several times, decreasing in the direction of crack propagation. Interstitial cracks (I) are characterized by a high frequency of *F_x_* changes while maintaining a similar *F_xmax_* value for each peak (see Figure 9). In this case, there is no sudden drop in the *F_x_* value. Continuous cracks, forming a continuous chip, generate a constant *F_x_* value for the longest cutting distance (see Figure 11). The analyses show a correlation between the nature of the crack and the value of the *F_x_* force.

The results, a detailed continuation of the research of Zawadzki [50], confirm the possibility of using abrasive tools for machining bone surfaces but require further analysis. The above experimental analyses can be reflected in the analyses presented in the introduction using the differential quadrature hierarchical finite element method or the Bezier method. The applied methods of analysis of composite structures, taking into account the direction of fiber distribution, can reflect the natural structure of osteons embedded in the interstitial matrix.

## 5. Conclusions

In this study, experiments were performed on the orthogonal cutting of the cortical bone. A wide range of cutting tool geometries and cut depths was used. The influence of cortical bone anisotropy and d on crack propagation, cutting force and crack mechanism was analyzed. The following conclusions can be drawn from the experimental results:The most advantageous and predictable processes were obtained for positive rake angles and a depth cut below 100 µm.For negative rake angle values, depending on the depth of cut, the following were distinguished: no cutting (friction), brittle deformation and shear cracks. The following was obtained for positive rake angle values: slight penetrating, interstitial cracks with continuous chip formation and horizontal and transverse cracks. For the zero rake angle, shear cracks predominate.There are regular cracks along the cement line in the parallel cutting direction. In the across direction, the cracks also spread along the cement line, although they were shorter and less regular, and they strongly propagated deeper into the material. Shear fractures concerning the osteon predominate in the transverse cutting direction.Crack propagation occurs along the cement line for the across and parallel direction. In the case of the transverse direction, stresses accumulate in the osteons, followed by uncontrolled fracture.The morphological parameters of the surface maintain a clear correlation with the cutting direction. Cutting in the parallel direction provides a surface with the lowest volume damage and roughness.The clearance angles do not affect the crack propagation character.

Based on the above results, the authors suggest not cutting the cortical bone in the direction transverse to the axis of osteon distribution during the cutting of the cortical bone. The optimal cutting depth is 50 to 100 μm, ensuring stable tissue breakage with the most efficient material removal. It is suggested to use a rake angle in the range of 0–20°. However, it is possible to use grinding wheels where the grains have negative rake angles, and the depth of cut must be optimized. The authors plan to develop experimental research with simulations related to machining with a single abrasive grain and then with a grain assembly. This research covered an extensive range of input parameters, which complicated the analysis of the results. In further research, the authors assume a reduction in the input data ranges to have a closer look at the results. The plans assume the application of numerical methods such as DQFEM, 2D-FEM and the Bezier method to develop a bone-tissue-cutting model.

## Figures and Tables

**Figure 1 jfb-13-00154-f001:**
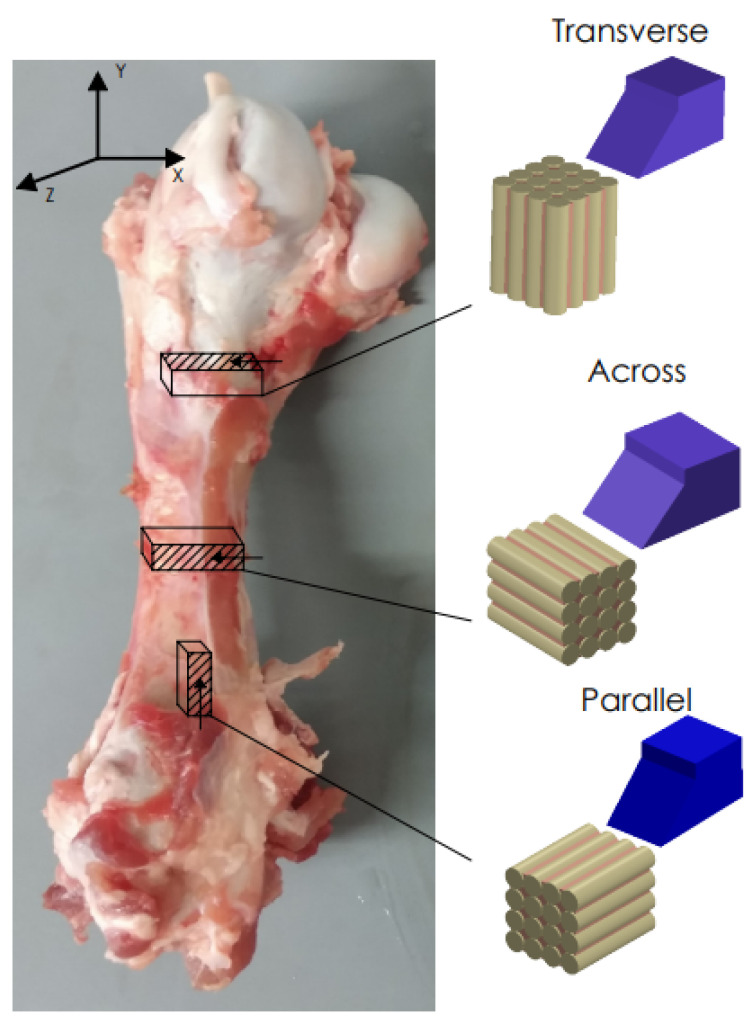
Preparation of bone specimens for transverse, across and parallel directions of osteons orientation.

**Figure 2 jfb-13-00154-f002:**
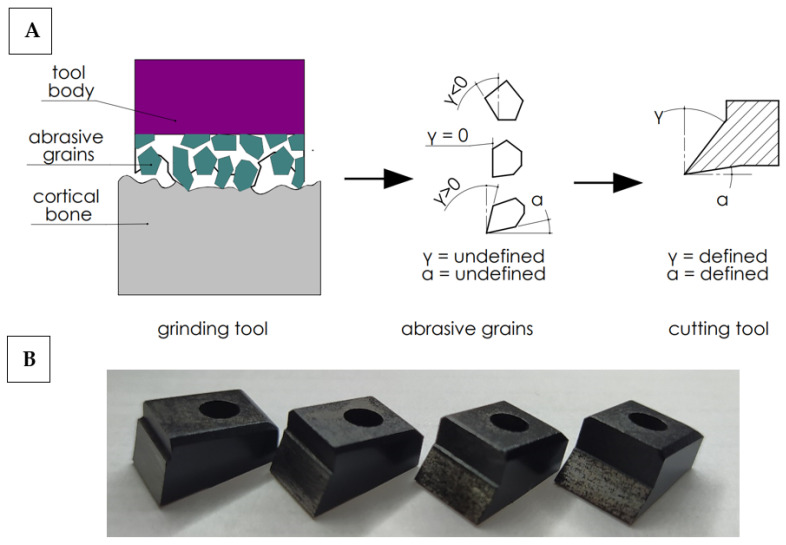
Diagram of simplified cutting geometry with an abrasive tool (**A**) and cutting tools (**B**). Transfer of cutting from abrasive grains with undefined geometry to cutting tools with a defined geometry.

**Figure 3 jfb-13-00154-f003:**
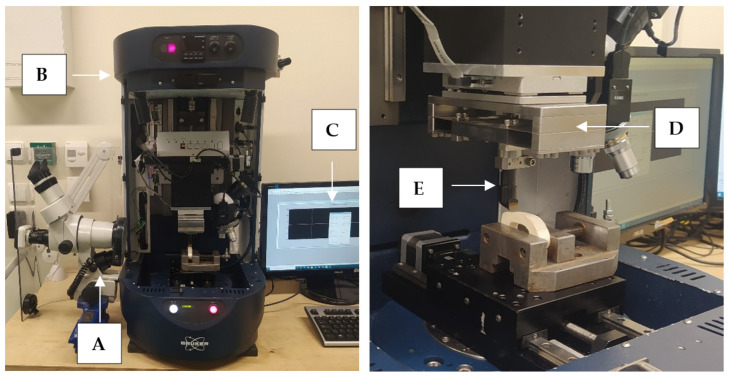
The measurement station: (**A**) optical microscope with a camera, (**B**) Tribo-tester Bruker UMT, (**C**) control and data recording system, (**D**) force sensor DFM−20, (**E**) cutting tool holder.

**Figure 4 jfb-13-00154-f004:**
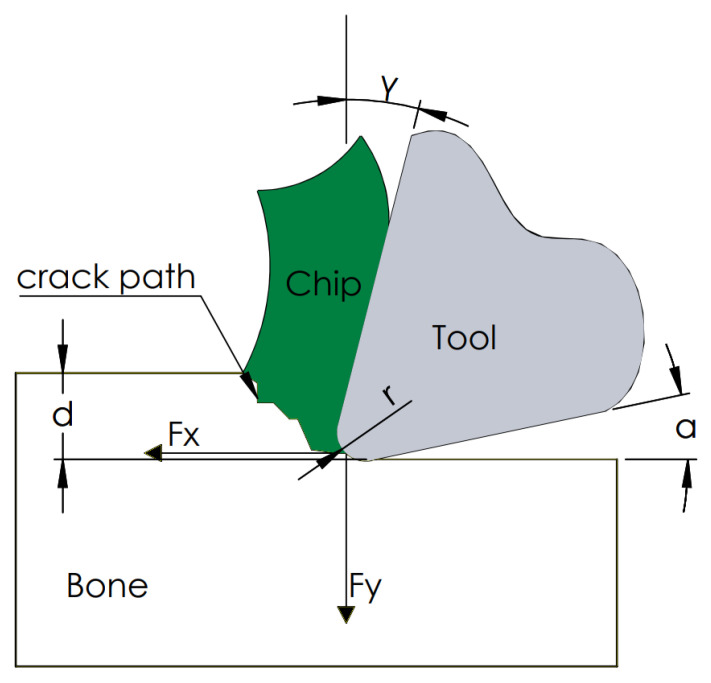
Schematic diagram of the used cutting model.

**Figure 5 jfb-13-00154-f005:**
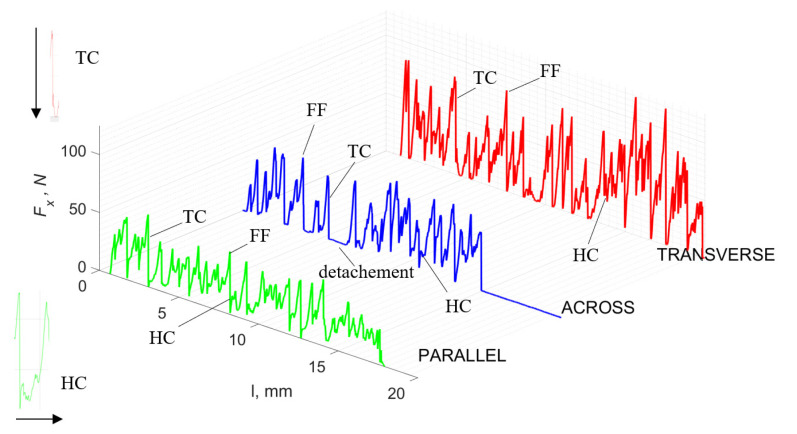
The course of changes in the value of the shear force *F_x_* concerning the path width *l* for three directions of osteon orientation: FF—fracture force, HC—horizontal cracks and TC—transverse cracks.

**Figure 6 jfb-13-00154-f006:**
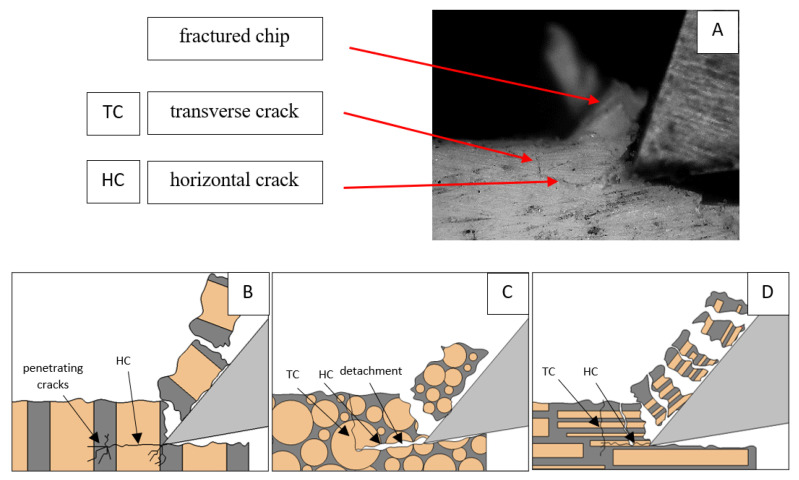
Graphics shows the process of cutting cortical bone tissue. Graphics (**A**): microscopic photography with an indication of chips, HC—horizontal cracks, TC—transverse cracks. Graphical representation of orthogonal cutting for individual osteon orientations: (**B**): transverse, (**C**): across and (**D**): parallel.

**Figure 7 jfb-13-00154-f007:**
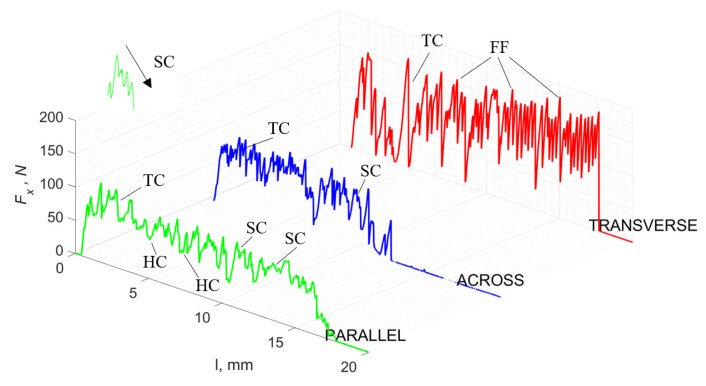
The course of changes in the value of the shear force *F_x_* concerning the path width *l* for three directions of osteon orientation. The symbols are marked: SC—shearing cracks, FF—fracture force, HC—horizontal cracks and TC—transverse cracks.

**Figure 8 jfb-13-00154-f008:**
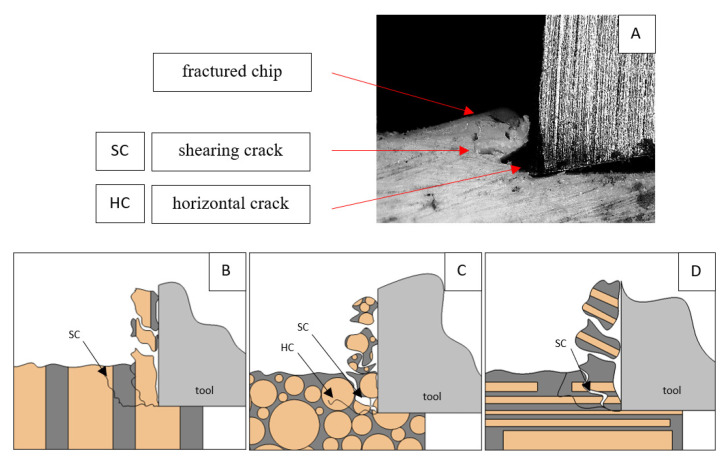
Graphics shows the process of cutting cortical bone tissue. Graphics (**A**): microscopic photography with an indication of chips, SC—shearing cracks, HC—horizontal crack. Graphical representation of orthogonal cutting for individual osteon orientation: (**B**): transverse, (**C**): across and (**D**): parallel.

**Figure 9 jfb-13-00154-f009:**
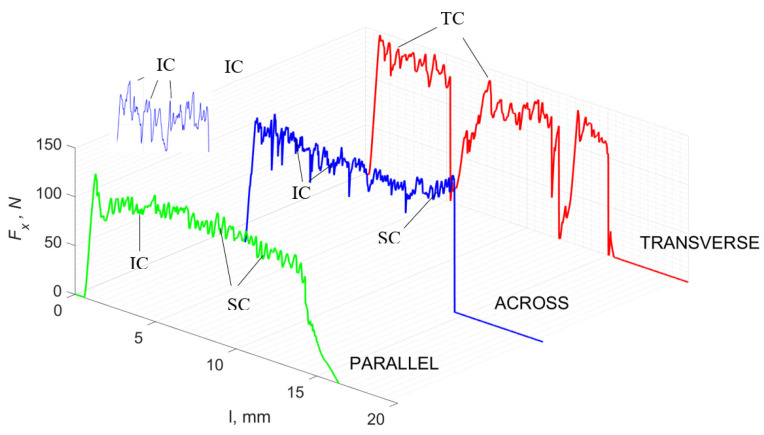
The course of changes in the value of the shear force *F_x_* concerning the path width *l* for three directions of osteon orientation. The symbols are marked: SC—shearing cracks, FF—fracture foce, IC—interstitial cracks and TC—transverse cracks.

**Figure 10 jfb-13-00154-f010:**
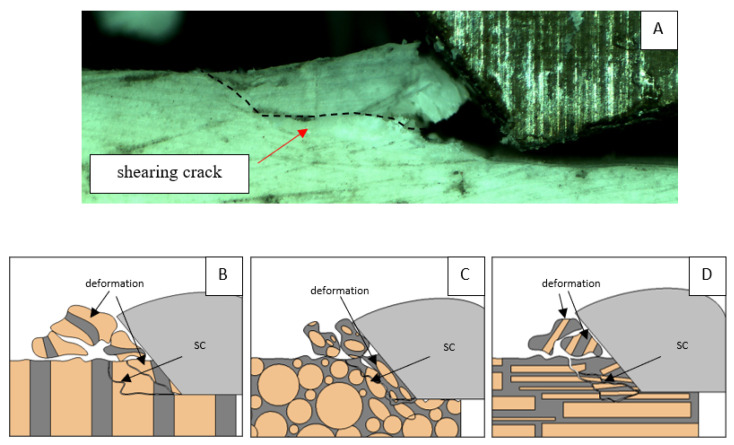
Graphics showing the process of cutting cortical bone tissue. Graphics (**A**): microscopic photography indicating chip and shearing crack. Graphical representation of orthogonal cutting for individual osteon orientations in the graphics: (**B**): transverse, (**C**): across and (**D**): parallel.

**Figure 11 jfb-13-00154-f011:**
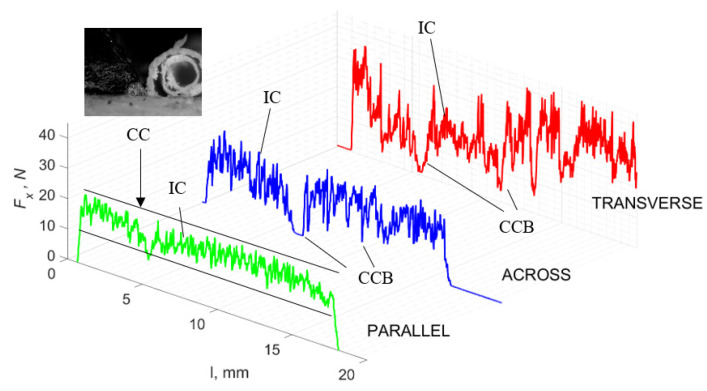
The course of changes in the value of the shear force *F_x_* concerning the path width *l* for three directions of osteon orientation. The symbols are IC—interstitial cracks CC—continuous chip and CCB—continous chip breakage.

**Figure 12 jfb-13-00154-f012:**
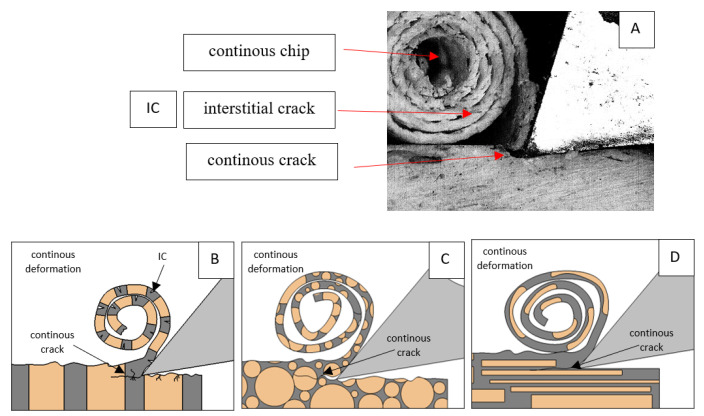
Graphics showing the process of cutting cortical bone tissue. Graphics (**A**): microscopic photography with an indication of chips, IC—interstitial cracks, CB—chip breaks. Graphical representation of orthogonal cutting for individual osteon orientations in the graphics: (**B**): transverse, (**C**): across and (**D**): parallel.

**Figure 13 jfb-13-00154-f013:**
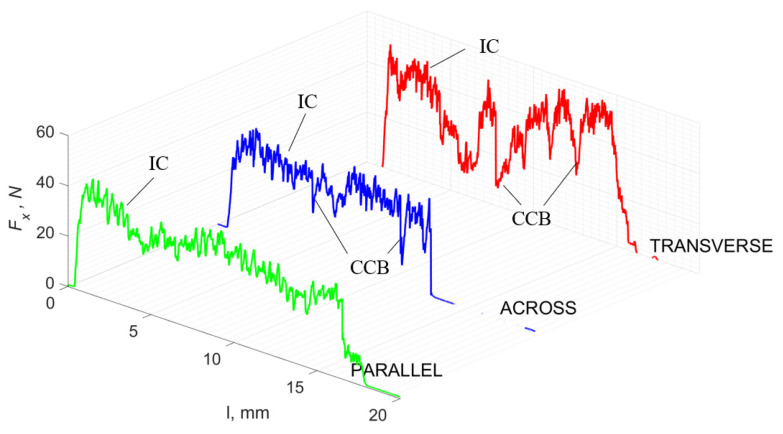
The course of changes in the value of the shear force *F_x_* concerning the path width *l* for three directions of osteon orientation. The symbols are IC—interstitial cracks and CCB—continuous chip break.

**Figure 14 jfb-13-00154-f014:**
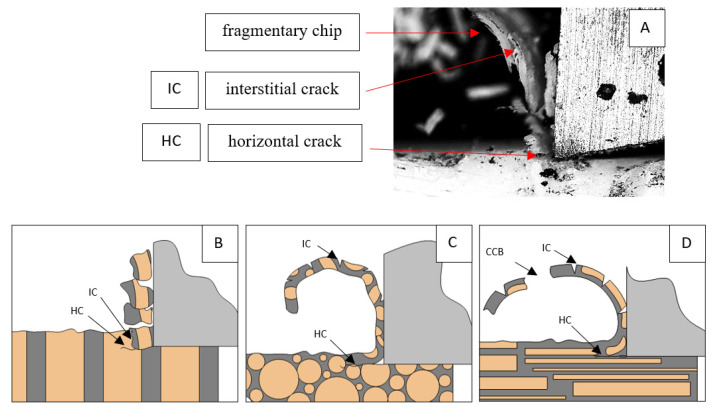
Graphics showing the process of cutting cortical bone tissue. Graphics (**A**): microscopic photography with an indication of chips, IC—interstitial cracks, CCB—continous chip break, HC—horizontal crack. Graphical representation of orthogonal cutting for individual osteon orientations in the graphics: (**B**): transverse, (**C**): across and (**D**): parallel.

**Figure 15 jfb-13-00154-f015:**
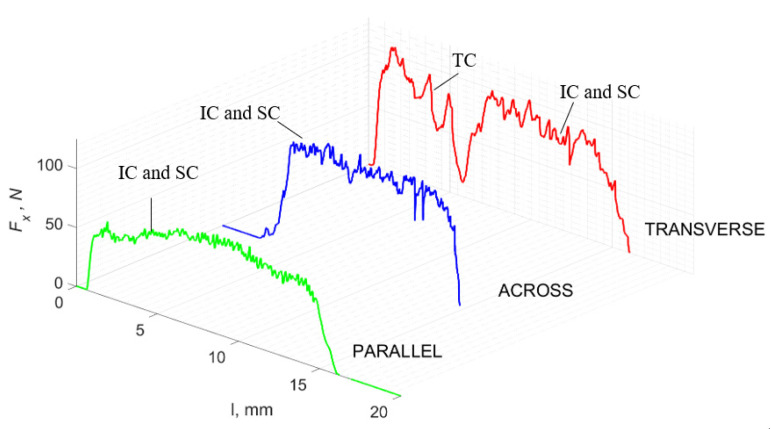
The course of changes in the value of the shear force *F_x_* concerning the path width *l* for three directions of osteon orientation. The symbols are SC—shear cracks and FB—fracture break.

**Figure 16 jfb-13-00154-f016:**
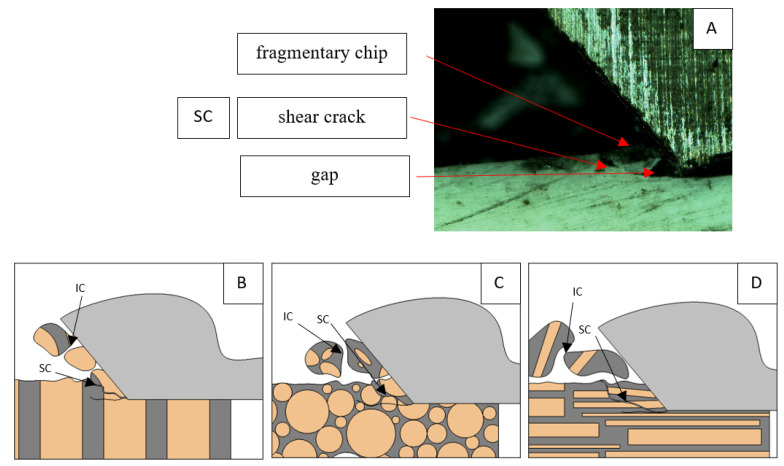
Graphics showing the process of cutting cortical bone tissue. Graphics (**A**): microscopic photography indicating chips, SC—shear cracks, IC—interstitial crack. Graphical representation of orthogonal cutting for individual osteon orientations in the graphics: (**B**): transverse, (**C**): across and (**D**): parallel.

**Figure 17 jfb-13-00154-f017:**
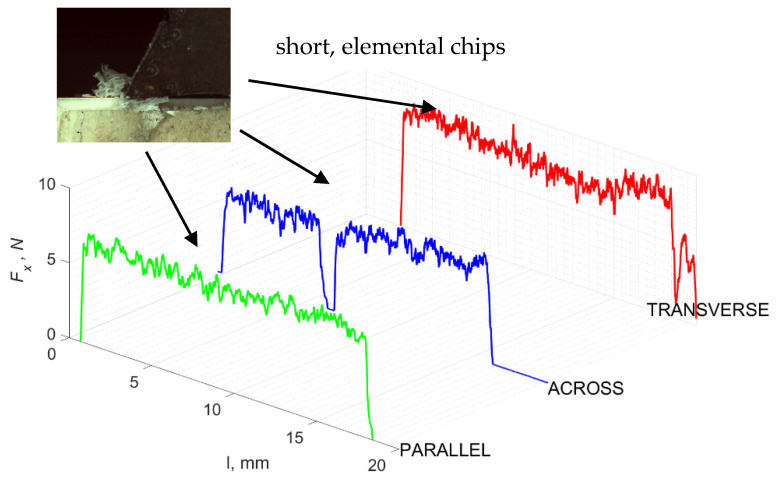
The course of changes in the value of the shear force *F_x_* concerning the path width *l* for three directions of osteon orientation.

**Figure 18 jfb-13-00154-f018:**
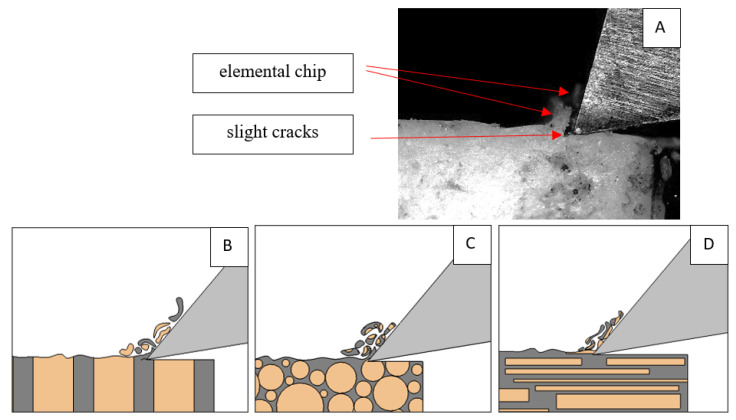
Graphics showing the process of cutting cortical bone tissue. Graphics (**A**): microscopic photography and representation of orthogonal cutting for individual osteon orientations in the graphics: (**B**): transverse, (**C**): across and (**D**): parallel.

**Figure 19 jfb-13-00154-f019:**
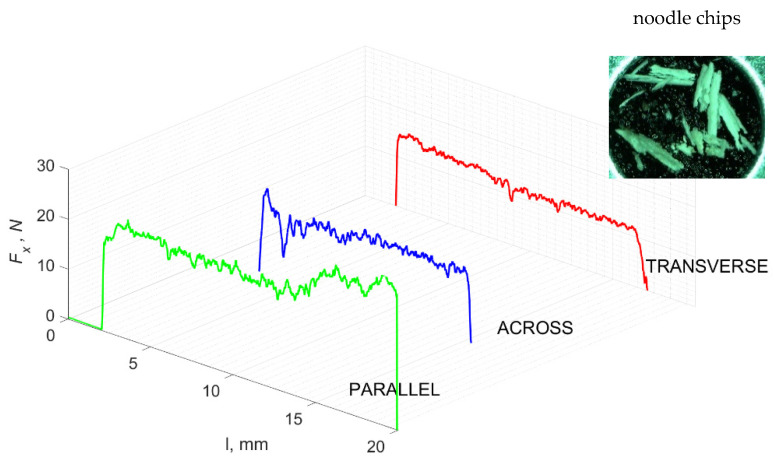
The course of changes in the value of the shear force *F_x_* concerning the path width *l* for three directions of osteon orientation.

**Figure 20 jfb-13-00154-f020:**
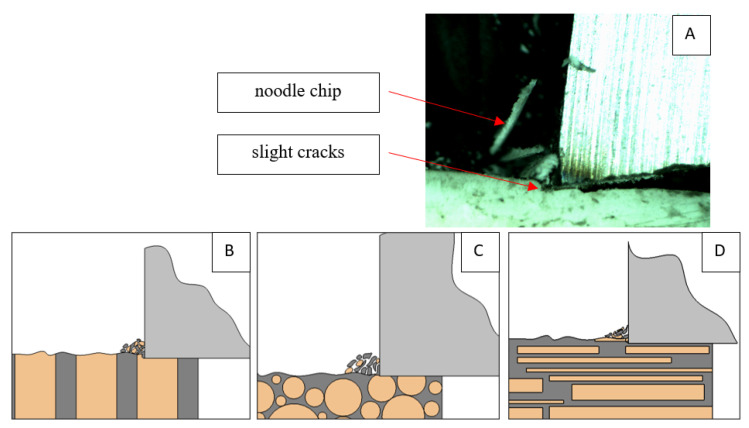
Graphics showing the process of cutting cortical bone tissue. Graphics (**A**): microscopic photography and representation of orthogonal cutting for individual osteon orientations in the graphics: (**B**): transverse, (**C**): across and (**D**): parallel.

**Figure 21 jfb-13-00154-f021:**
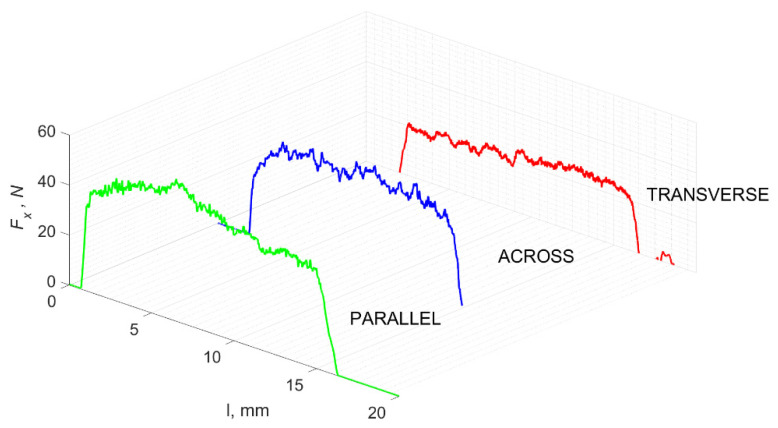
The course of changes in the value of the shear force *F_x_* concerning the path width *l* for three directions of osteon orientation.

**Figure 22 jfb-13-00154-f022:**
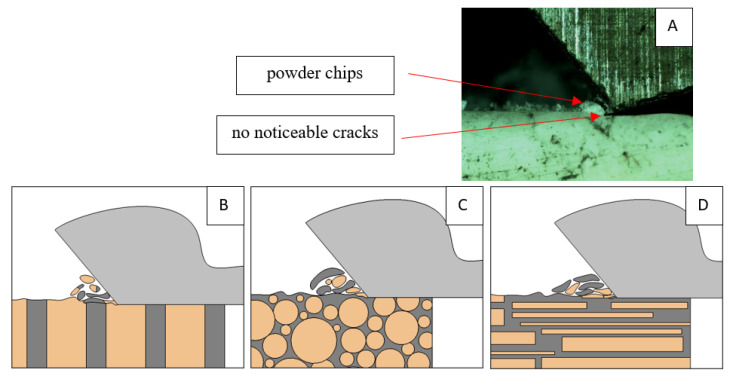
Graphics showing the process of cutting cortical bone tissue. Graphics (**A**): microscopic photography and representation of orthogonal cutting for individual osteon orientations in the graphics: (**B**): transverse, (**C**): across and (**D**): parallel.

**Figure 23 jfb-13-00154-f023:**
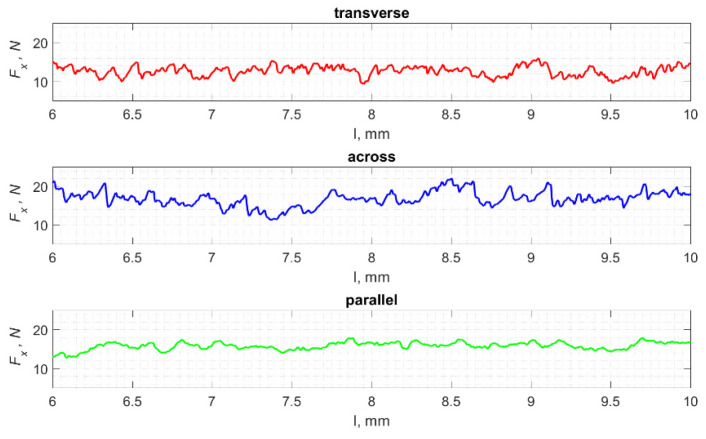
The specified course of changes in the value of *F_x_* as a function of path *l*, for three cutting directions.

**Figure 24 jfb-13-00154-f024:**
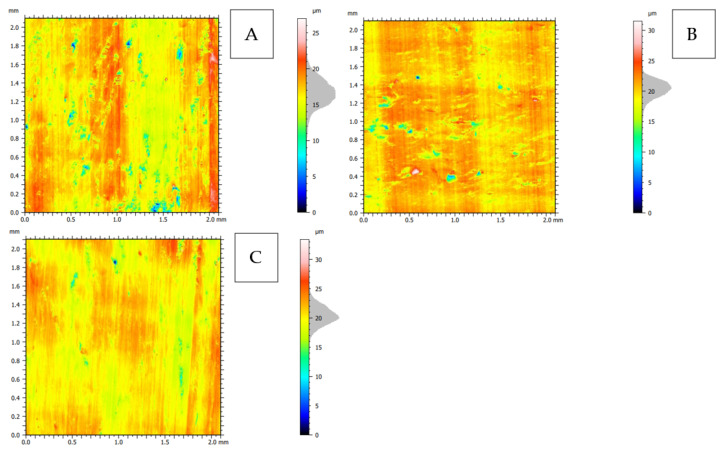
2D topography representation of the surface after orthogonal cutting in the direction: (**A**): transverse, (**B**): across and (**C**): parallel.

**Table 1 jfb-13-00154-t001:** Base experiment parameters.

Parameter	Nomenclature and Unit	Value
Rake angle	*γ*, °	40, 30, 20, 10, 0, −10, −20, −30, −40
Clearance angle	*α*, °	5, 10, 15
Cutting depth	*d*, μm	0.5, 1, 2, 2.5, 5, 10, 25, 50, 100, 125, 150, 175
Cutting velocity	*v_c_*, mm/min	30

**Table 2 jfb-13-00154-t002:** Surface morphology parameters depend on the cutting direction.

Parameter	Direction		
	Transverse	Across	Parallel
*Vmp*, mL/m^2^	0.1	0.118	0.0589
*Vmc*, mL/m^2^	2.56	1.71	0.919
*Vvc*, mL/m^2^	3.29	2.17	1.29
*Vvv*, mL/m^2^	0.715	0.571	0.185
*Iz*, %	17.7	34.4	32.3
*d_f_*, μm	5.22	3.74	1.88
*Rq*, μm	1.53	1.18	1.1

**Table 3 jfb-13-00154-t003:** Crack types were observed during orthogonal cortical bone cutting analyses. Crack marks according to the legend: T—transverse, H—horizontal, S—shear, I—interstitial, C—continous, N—no visible, SP—slight penetrating, D—deformation of material and F—surface rubbing.

*γ*, *°*	*d*, μm										
	0.5	1	2.5	5	10	25	50	100	125	150	175
	TRANSVERSE										
−40	N, F	N, F	N, F	F, B, D	S, B, D	S, B, D	S, B, D	S, B, D	S, B, D	S, B, D	S, B, D
−30	N, F	N, F	N, F	F, B, D	F, B, D	S, B, D	S, B, D	S, B, D	S, B, D	S, B, D	S, B, D
−20	N, F	N, F	N, F	F, B, D	S, B, D	S, B, D	S, B, D	S, B, D	S, B, D	S, B, D	S, B, D
−10	N, F	N, F	SP, F	SP, F	S, B, D	S, B, D	S, B, D	S, B, D	S, B, D	S, B, D	S, B, D
0	SP, F	SP, F	SP, F	S, C, I	S, C, I	S, C, I	S, C, I	S, C, I	S, B, D	S, B, D	S, B, D
10	SP	SP	SP	I, C	I, C	I, C	I, C	T, H	T, H	T, H	T, H
20	SP	SP	SP	I, C	I, C	I, C	I, C	T, H, C	T, H	T, H	T, H
30	SP	SP	SP	I, C	I, C	I, C	I, C	T, H, C	T, H	T, H	T, H
40	SP	SP	SP	I, C	I, C	I, C	I, C	T, H, C	T, H	T, H	T, H
***γ*, *°***	**PARALLEL**										
−40	N, F	N, F	N, F	F, B, D	S, B, D	S, B, D	S, B, D	S, B, D	S, B, D	S, B, D	S, B, D
−30	N, F	N, F	N, F	F, B, D	S, B, D	S, B, D	S, B, D	S, B, D	S, B, D	S, B, D	S, B, D
−20	N, F	N, F	N, F	F, B, D	S, B, D	S, B, D	S, B, D	S, B, D	S, B, D	S, B, D	S, B, D
−10	N, F	SP, F	SP, F	SP, F	S, B, D	S, B, D	S, B, D	S, B, D	S, B, D	S, B, D	S, B, D
0	SP, F	SP, F	SP, F	I, C	I, C	I, C	S, I, C	S, I, C	S, B, D	S, B, D	S, B, D
10	SP	SP	SP	I, C	I, C	I, C	I, C	T, H	T, H	T, H	T, H
20	SP	SP	SP	I, C	I, C	I, C	I, C	T, H, C	T, H	T, H	T, H
30	SP	SP	SP	I, C	I, C	I, C	I, C	T, H, C	T, H	T, H	T, H
40	SP	SP	SP	I, C	I, C	I, C	I, C	T, H, C	T, H	T, H	T, H
***γ*, *°***	**ACROSS**										
−40	N, F	N, F	N, F	F, B, D	S, B, D	S, B, D	S, B, D	S, B, D	S, B, D	S, B, D	S, B, D
−30	N, F	N, F	N, F	F, B, D	S, B, D	S, B, D	S, B, D	S, B, D	S, B, D	S, B, D	S, B, D
−20	N, F	N, F	N, F	F, B, D	S, B, D	S, B, D	S, B, D	S, B, D	S, B, D	S, B, D	S, B, D
−10	N, F	N, F	SP, F	SP, F	S, B, D	S, B, D	S, B, D	S, B, D	S, B, D	S, B, D	S, B, D
0	SP, F	SP, F	SP, F	I, C	S, I, C	S, I, C	S, I, C	S, I, C	S, B, D	S, B, D	T, H
10	SP	SP	SP	I, C	I, C	I, C	I, C	T, H	T, H	T, H	T, H
20	SP	SP	SP	I, C	I, C	I, C	I, C	T, H, C	T, H	T, H	T, H
30	SP	SP	SP	I, C	I, C	I, C	I, C	T, H, C	T, H	T, H	T, H
40	SP	SP	SP	I, C	I, C	I, C	I, C	T, H, C	T, H	T, H	T, H

## Data Availability

Not applicable.

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
