# Peer review of "Bone Abrasive Machining: Influence of Tool Geometry and Cortical Bone Anisotropic Structure on Crack Propagation"

_jfb, 2022, doi:10.3390/jfb13030154_

Round 1
Reviewer 1 Report
I reviewed the article "Bone abrasive machining: influence of tool geometry and cortical bone anisotropic structure on crack propagation: and found it can be accepted after major revision.
-Describe the amount of force applied by the cutting tool. What cutting machine is used?
-How much was considered the compressive strength of the bone?
-What is the relationship between porosity and cutting quality?
-Are the authors trying to perform experiments on artificial scaffolds that other researchers have worked on?
-Abaqus evaluation models were reviewed.
-Table 1 needs to be repainted and simplified.
-The language of the article should be improved
-The overall conclusion should be improved and the objectives and results should be clearly stated. A review of the previous materials and the achievements of other scholars has been done
-Hashemi, S., Esmaeili, S., Ghadirinejad, M., Saber-Samandari, S., Sheikhbahai, E., Kordjamshidi, A., Khandan, A. (2020). Micro-Finite Element Model to Investigate the Mechanical Stimuli in Scaffolds Fabricated via Space Holder Technique for Cancellous Bone. ADMT Journal, 13(1), 51-58.
Author Response
Dear Reviewer,
Thank you very much for the review of the article and the possibility of making corrections. Below I provide information on the changes made to the comments:
- Abstract has been thoroughly rebuilt according to the reviewer's indications. Detailed results and conclusions were introduced.
- The description of the cutting tool geometry has been extended (see line 163).
- Unfortunately, due to the biological nature of the samples, it was partially disposed of. However, the literature data on elemental composition were introduced into the article (see line 125).
- Three clearance angles were used in the studies, although this angles did not affect crack propagation (see line 219).
- Coordinates have been added to Figure 1. The font in all Figures has been enlarged.
- In Figure 10a, a better-quality photograph was introduced. The description of the phenomenon has been extended (see line 309).
- There was a discussion with the literature dealing with the issues of the "Differential Quadrature" and "Bezier" methods in both introduction (see line 80) and discussion (555).
- Future research directions are included in the Conclusions (see line 588), and limitations are also indicated (see line 590).
- The linguistic and stylistic correction was also made. The author would like to point out that if the language of the article still requires corrections, he will direct it to English language editing arranged by MDPI.
Additionally, following the comments of other reviewers, the following changes were introduced:
- Discussions on the influence of porosity on crack treatment and propagation were presented (see lines 91 and 449).
- Bone samples were collected from mature specimens. The cortical bone, which has the most homogeneous and unchanging properties, was selected for the study. This way, possible osteoporosis's influence on the tissue's structure was avoided.
I hope I met the suggested requirements. I am ready to introduce further amendments.
Yours faithfully,
Paweł Zawadzki
Reviewer 2 Report
Please read and fully address the comments listed below:
1- The ABSTRACT is not written in a logical order. Start with an overview of the topic and a rationale for your paper. Describe the methodology you used and the general outline of the manuscript. Also, in the end, state the result in more detail (i.e., provide some numbers).
2- Section 2.3 (cutting tools geometry) needs more explanation.
3- Please further characterize the bone powder using XRF (find elemental composition of TPU powder) and XRD (find phase identification of a crystalline material) tests.
4- In Table 1, please justify your choice of the selected clearance angles.
5- Add XYZ coordinates to Figure 1 and all other figures. Increase the font size in all figures (e.g., Figs, 19, and 21)
6- In Figure 10-a, please provide more close-up images of the cracked planes, and further discuss the modes of failure.
3. Crack analysis of anisotropic media is experimentally explored in your paper, which can also be done using XFEM as well. However, among many numerical techniques, the “Differential Quadrature” and “Bezier” methods proved to have higher stability and accuracy than other numerical methods (e.g., XFEM). For this purpose, please write a paragraph in your paper introducing these methods which can “alternatively” solve your problem and reference the landmark papers listed below (selected from Dr. Liu and Dr. Aghdam's research labs):
Differential Quadrature Method:
• Wu, Y., Xing, Y., & Liu, B. (2018). Analysis of isotropic and composite laminated plates and shells using a differential quadrature hierarchical finite element method. Composite Structures, 205, 11-25.
• Yan, Y., Liu, B., Xing, Y., Carrera, E., & Pagani, A. (2021). Free vibration analysis of variable stiffness composite laminated beams and plates by novel hierarchical differential quadrature finite elements. Composite Structures, 274, 114364.
Bezier Method
• Kabir, H., & Aghdam, M. M. (2019). A robust Bézier based solution for nonlinear vibration and post-buckling of random checkerboard graphene nano-platelets reinforced composite beams. Composite Structures, 212, 184-198.
• Kabir, H., & Aghdam, M. M. (2021). A generalized 2D Bézier-based solution for stress analysis of notched epoxy resin plates reinforced with graphene nanoplatelets. Thin-Walled Structures, 169, 108484.
7- Conclusion: Can authors highlight future research directions and recommendations? Also, highlight the assumptions and limitations (e.g 1-2 shortcoming(s) of the present study)? Besides, recheck your manuscript and polish it for grammatical mistakes (you can use “Grammarly to quickly edit your document).
Author Response
Dear Reviewer,
Thank you very much for the review of the article and the possibility of making corrections. Below I provide information on the changes made to the comments:
- Abstract has been thoroughly rebuilt according to the reviewer's indications. Detailed results and conclusions were introduced.
- The description of the cutting tool geometry has been extended (see line 163).
- Unfortunately, due to the biological nature of the samples, it was partially disposed of. However, the literature data on elemental composition were introduced into the article (see line 125).
- Three clearance angles were used in the studies, although these angles did not affect crack propagation (see line 219).
- Coordinates have been added to Figure 1. The font in all Figures has been enlarged.
- In Figure 10a, a better-quality photograph was introduced. The description of the phenomenon has been extended (see line 309).
- There was a discussion with the literature dealing with the issues of the "Differential Quadrature" and "Bezier" methods in both introduction (see line 80) and discussion (555).
- Future research directions are included in the Conclusions (see line 588), and limitations are also indicated (see line 590).
- The linguistic and stylistic correction was also made. The author would like to point out that if the language of the article still requires corrections, he will direct it to English language editing arranged by MDPI.
Additionally, following the comments of other reviewers, the following changes were introduced:
- Discussions on the influence of porosity on crack treatment and propagation were presented (see lines 91 and 449).
- Bone samples were collected from mature specimens. The cortical bone, which has the most homogeneous and unchanging properties, was selected for the study. This way, possible osteoporosis's influence on the tissue's structure was avoided.
I hope I met the suggested requirements. I am ready to introduce further amendments.
Yours faithfully,
Paweł Zawadzki
Reviewer 3 Report
The present study is fascinating however some minor changes are required.
-Throughout the manuscript, there are several typos that must be corrected like continuous, characterized, etc.
-Was there some correlation between the force value and the number or type of cracks?
-The bone samples were taken from a young specimen or an adult one. Age affects bone structure.
-Other parameters like bone porosity may affect also crack propagation which should be at least discussed.
Author Response
Dear Reviewer.
Thank you very much for the review of the article and the possibility of making corrections. Below I provide information on the changes made to the comments:
- Typos, incorrect phrases and symbols were corrected.
- The force-crack correlation was described in the discussion (see line 540).
- Bone samples were collected from mature specimens. The cortical bone, which has the most homogeneous and unchanging properties, was selected for the study. This way, possible osteoporosis's influence on the tissue's structure was avoided.
- The potential influence of porosity on the crack propagation and the cutting process has been added and described (see lines 114 and 454)
- The linguistic and stylistic correction was also made. The author would like to point out that if the language of the article still requires corrections, he will direct him to English language editing arranged by MDPI.
Additionally, following the comments of other reviewers, the following changes were introduced:
- Future research directions are included in Conclusions (see line 588) and limitations are also indicated (see line 590)
- Discussions on the influence of porosity on crack treatment and propagation were presented (see line 91 and 449)
- The description of the cutting-edge geometry has been extended (see line 163)
- Unfortunately, due to the biological nature of the samples, they were partially disposed of. However, the literature data on elemental composition were introduced into the article (see line 125).
- Coordinates have been added to Figure 1. The font in all Figures has been enlarged.
- In Figure 10a, a better-quality photograph is introduced. The description of the phenomenon has been extended (see line 309)
- There was a discussion with the literature dealing with the issues of the "Differential Quadrature" and "Bezier" methods in both introduction (see line 80) and discussion (555).
- Future research directions are included in Conclusions (see line 588) and limitations are also indicated (see line 590)
I hope, I met the suggested requirements. I am ready to introduce further amendments.
Yours faithfully,
Paweł Zawadzki
Reviewer 4 Report
The manuscript “Bone abrasive machining: influence of tool geometry and cortical bone anisotropic structure on crack propagation” by Zawadzki and Talar reports the impact of grain geometry and the anisotropic structure of bone tissue on the cutting process and crack propagation. Although, the experimental was well carried out, the presentation should be reorganized. Therefore, I would suggest authors may take a least a major revision of this work. Here are the comments and suggestions:
1. The abstract and conclusions should be revised.
2. This work seems more like an experimental report than a paper. Some results can be moved to the supporting materials.
Author Response
Dear Reviewer.
Thank you very much for the review of the article and the possibility of making corrections. Below I provide information on the changes made to the comments:
- Abstract and Conclusions have been thoroughly rebuilt to show the essential elements of the work.
- Numerous changes were introduced to increase the scientific nature of the work, as presented below.
- The linguistic and stylistic correction was also made. The author would like to point out that if the language of the article still requires corrections, he will direct him to English language editing arranged by MDPI.
Additionally, following the comments of other reviewers, the following changes were introduced:
- Future research directions are included in Conclusions (see line 588) and limitations are also indicated (see line 590)
- Discussions on the influence of porosity on crack treatment and propagation were presented (see lines 91 and 449)
- The description of the cutting tool geometry has been extended (see line 163)
- Unfortunately, due to the biological nature of the samples, they were partially disposed of. However, the literature data on elemental composition were introduced into the article (see line 125).
- Coordinates have been added to Figure 1. The font in all Figures has been enlarged.
- In Figure 10a, a better-quality photograph is introduced. The description of the phenomenon has been extended (see line 309)
- There was a discussion with the literature dealing with the issues of the "Differential Quadrature" and "Bezier" methods in both introduction (see line 80) and discussion (555).
- Future research directions are included in Conclusions (see line 588) and limitations are also indicated (see line 590)
- Bone samples were collected from mature specimens. The cortical bone, which has the most homogeneous and unchanging properties, was selected for the study. This way, possible osteoporosis's influence on the tissue's structure was avoided.
I hope, I met the suggested requirements. I am ready to introduce further amendments.
Yours faithfully,
Paweł Zawadzki
Round 2
Reviewer 1 Report
The paper is well revised and can be accepted in this form.
Reviewer 2 Report
The authors addressed my comments and the manuscript can be published in the present format.
Reviewer 4 Report
It seems more acceptable now.